# Autoimmune Pancreatitis Mimicking a Pancreatic Neuroendocrine Tumor: A Case Report with a Literature Review

**DOI:** 10.3390/ijms26041536

**Published:** 2025-02-12

**Authors:** Marianna Franchina, Liliana Dell’Oro, Sara Massironi

**Affiliations:** 1Division of Gastroenterology, ASST Papa Giovanni XXIII, 24127 Bergamo, Italy; mfranchina@asst-pg23.it; 2Department of Medicine and Surgery, University of Milano-Bicocca, 20126 Milan, Italy; l.delloro2@campus.unimib.it; 3Department of Medicine and Surgery, Vita-Salute San Raffaele University, 20132 Milan, Italy; 4Istituti Ospedalieri Bergamaschi, 24040 Zingonia, Italy

**Keywords:** autoimmune pancreatitis, chronic pancreatitis, pancreatic neuroendocrine tumors

## Abstract

Autoimmune pancreatitis (AIP) is a rare chronic pancreatitis subtype that often mimics pancreatic cancer due to the overlapping clinical and radiological features, posing significant diagnostic challenges. Similarly, distinguishing AIP from pancreatic neuroendocrine neoplasms (PanNENs), which present with nonspecific symptoms, adds complexity to clinical evaluations. We present the case of a 46-year-old male with recurrent acute idiopathic pancreatitis. Abdominal computed tomography (CT) revealed a 25 mm hypodense mass in the pancreatic tail with mild arterial contrast enhancement. Magnetic resonance imaging (MRI) showed the mass to be hypointense on T2-weighted sequences, with no diffusion restriction and an enhancement pattern akin to normal pancreatic tissue. The endoscopic ultrasound-guided fine needle biopsy (EUS-FNB) was inconclusive. Gallium-68 DOTATATE positron emission tomography–CT (Ga-68 DOTATATE PET-CT) showed an increased tracer uptake, leading to a distal pancreatectomy with a splenectomy. Histopathology demonstrated chronic sclerotic pancreatitis with inflammatory infiltrates. Elevated serum IgG4 levels confirmed the diagnosis of type 1 AIP Differentiating AIP from pancreatic malignancies, including PanNENs, is both critical and complex. This case highlights a misdiagnosis of PanNENs in a patient with focal AIP, where neuroendocrine hyperplasia and islet cell clusters within fibrotic areas mimicked PanNENs, even on Ga-68 PET-CT. The findings emphasize the potential for false positives with Ga-68 DOTATATE PET-CT and the importance of integrating clinical, radiological, and histological data for an accurate diagnosis.

## 1. Introduction

Autoimmune pancreatitis (AIP) is a rare form of chronic pancreatitis characterized by obstructive jaundice, with or without pancreatic masses, lymphoplasmacytic infiltration, fibrosis, and a remarkable response to steroids [1]. While considered uncommon, its exact prevalence and incidence remain unknown. Increased awareness and the development of diagnostic criteria in recent years have significantly improved the recognition of AIP [2].

There are two distinct types of AIP, each differing in epidemiology, pathogenesis, histology, and natural history. Type 1 AIP, also known as lymphoplasmacytic sclerosing pancreatitis, is the pancreatic manifestation of an IgG4-related disease—a systemic fibroinflammatory condition characterized by chronic, relapsing courses. It is the more frequent form of AIP, predominantly affecting middle-aged men. The clinical symptoms include jaundice and weight loss, typically without abdominal pain. It is often associated with other organ involvement and is histologically characterized by IgG4-positive plasma cell infiltration, storiform fibrosis, and obliterative phlebitis [3]. Type 1 AIP frequently recurs after steroid withdrawal [3,4].

Type 2 AIP, also called idiopathic duct-centric pancreatitis, affects both genders and typically younger individuals. It is often associated with inflammatory bowel disease (IBD) in about 16–30% of cases, especially ulcerative colitis [4]. It usually presents as acute pancreatitis and is histologically characterized by granulocytic epithelial lesions. Unlike Type 1, it generally does not recur after stopping treatment [4,5].

Since 2002, several diagnostic criteria have been proposed to aid clinicians in diagnosing AIP [3,6,7,8].

The International Consensus Diagnostic Criteria (ICDC), which considers imaging features, other organ involvement, histology, and treatment response, are the most widely used guidelines. These criteria differentiate between Type 1 and Type 2 AIP and assign varying levels of evidence to each diagnostic criterion [1].

Type 1 AIP is diagnosed based on a combination of clinical, imaging, and serological criteria, including elevated serum IgG4 levels and characteristic radiological features such as diffuse or focal pancreatic thickening and a peripancreatic “halo” [9]. The diagnosis of Type 2 AIP is based on histology, the presence of the associated IBD, the absence of elevated serum IgG4 levels, and a rapid response to steroid treatment [10].

Pancreatic neuroendocrine tumors (PanNENs) are a heterogeneous group of neoplasms originating from neuroendocrine cells [11,12]. Although rare—with a reported incidence of 0.48 per 100,000—they represent 1–2% of pancreatic neoplasms, and their incidence has significantly increased over recent decades [13,14,15]. Indeed, they are the second most common solid pancreatic neoplasm [16,17]. These tumors typically affect individuals around 55 years of age, with no significant gender predilection. Most are sporadic, but about 1–2% of all PanNENs, and specifically 17% of the functioning forms, are associated with familial syndromes such as multiple endocrine neoplasia type 1, tuberous sclerosis, von Hippel–Lindau syndrome, and neurofibromatosis type 1 [18,19].

PanNENs are classified as functioning or non-functioning based on hormone secretion. Non-functioning PanNENs, which account for 50–85% of cases, are often diagnosed incidentally through advanced imaging and are generally asymptomatic or present with nonspecific symptoms such as abdominal pain and weight loss due to mass effect. Functioning PanNENs, though less common, include insulinomas, gastrinomas, glucagonomas, VIPomas, and somatostatinomas, each associated with distinct clinical syndromes [18,20].

The wide variability of PanNEN symptoms makes their diagnosis challenging, often leading to delays [20]. The cornerstones of a PanNEN diagnosis are radiological techniques such as computed tomography (CT) and magnetic resonance imaging (MRI), functional imaging—especially Gallium-68 DOTATATE positron emission tomography–CT (Ga-68 DOTATATE PET-CT)—and endoscopic ultrasonography (EUS) with tissue sampling [15,18,21,22]. CT and MRI with contrast enhancement are usually the initial imaging studies and typically identify PanNENs as circumscribed hypervascular solid masses, with features varying depending on tumor size [23,24,25]. Ga-68 DOTATATE PET-CT, which uses a radiolabeled tracer that binds to the somatostatin receptors on neuroendocrine cells, is invaluable for detecting small or obscure PanNENs. It aids in diagnosing functioning PanNENs, differentiating pancreatic masses, and identifying metastatic disease or recurrences [26,27].

EUS is the most accurate technique for detecting small PanNENs and offers the additional advantage of obtaining biopsies for diagnosis [22,28]. General circulating biomarkers, such as chromogranin A, can be monitored throughout the course of the disease, while specific biomarkers like insulin, gastrin, glucagon, VIP, and somatostatin levels are measured in cases of suspected functioning PanNENs [15,18]. The management of PanNENs depends on tumor size, the presence of metastasis, and hormone secretion, ranging from active surveillance to medical treatment, surgery, and peptide receptor radionuclide therapy [15,18].

In this report, we present the case of a 46-year-old male patient with focal AIP, initially misdiagnosed as PanNEN.

## 2. Case Presentation

A 46-year-old male presented to our Center for Pancreas Disease as an outpatient, reporting chronic abdominal pain suggestive of acute or recurrent pancreatitis or chronic pancreatitis. He denied alcohol consumption or smoking. The patient’s family history was notable for autoimmune conditions, including his father’s diagnosis of polymyositis and his sister’s diagnosis of fibromyalgia. His past medical history included metabolic syndrome (hypertension, obesity with a BMI of 33, and dyslipidemia), hepatic steatosis, gastroesophageal reflux disease (GERD) with a hiatal hernia, biliary sludge, and adenomyomatosis of the gallbladder fundus.

The patient’s pancreatic symptoms began in 2020, when he was diagnosed with acute pancreatitis at another center. The laboratory tests at that time revealed neutrophilic leukocytosis, C-reactive protein (CRP) levels elevated to three times the upper limit of normal (ULN), and moderate pancreatic enzyme elevation (amylase and lipase at 1.25 times ULN). Computed tomography (CT) showed peripancreatic fat stranding, multiple reactive lymph nodes located along the splenic artery and within the splenic hilum, and a thickening of the left anterior parenchymal fascia, consistent with acute pancreatitis. No evidence of cholelithiasis or biliary tract dilation was observed.

At a one-month follow-up CT scan, performed with iodinated contrast media according to a multiphase protocol, a more compact glandular component with polylobulated margins, was identified at the distal pancreatic tail. It measured 23 mm × 18 mm and was isodense compared to the surrounding pancreatic parenchyma during all phases (Figure 1).

Further evaluation using Gallium-68 DOTATATE positron emission tomography–CT (Ga-68 DOTATATE PET-CT) revealed an area of increased uptake in the distal pancreatic tail (SUV max 19), suggestive of a lesion expressing somatostatin receptors (Figure 2).

General blood neuroendocrine marker levels, including chromogranin A and neuron-specific enolase, were within normal limits, as were specific markers such as 5-hydroxyindoleacetic acid (5-HIAA), glucagon, insulin, and gastrin levels.

The endoscopic ultrasound (EUS) identified a 10.2 mm hypoechogenic oval lesion without a Doppler signal. However, the fine needle biopsy (FNB) result was insufficient for a diagnosis, due to limited cellularity and difficulty with cytologic interpretation.

In June 2021, the patient experienced another episode of epigastric pain that required hospitalization at another center. This was followed by a similar episode in February 2022, for which the patient was admitted to our center. The laboratory investigations during these episodes revealed elevated inflammatory markers, with CRP levels reaching four times the upper limit of normal (ULN), while pancreatic enzyme levels remained near-normal at 1.5 times the ULN.

Magnetic resonance imaging (MRI) conducted during this period confirmed the presence of a pseudonodular alteration in the pancreatic tail, measuring approximately 25 mm. The lesion exhibited a minimally hypointense signal on the T2-weighted sequences, no diffusion restriction, and mild enhancement on the arterial-phase. No pancreatic duct nor biliary tract dilation was observed (Figure 3).

Serological testing revealed elevated IgG4 levels at 1084 mg/dL (normal range: 39–864), while the total IgG levels were within normal limits at 4610 mg/dL. Based on these findings, the patient was discharged in good condition and scheduled for an outpatient follow-up.

Taking into account the Ga-68 DOTATATE PET-CT findings, which demonstrated the slow-growth lesion with somatostatin receptor expression, the pseudonodular alteration was interpreted as a neuroendocrine neoplasm.

Due to the high clinical and radiological suspicion, the likelihood of a subsequent surgical intervention, and the patient’s preference to avoid further invasive procedures, the FNB was not repeated. After discussing therapeutic options with the patient, a video-laparoscopic distal pancreatectomy with a splenectomy was performed in March 2022. Although a pancreatectomy without a splenectomy might have sufficed for a pancreatic neuroendocrine neoplasm, it was not pursued due to the lesion’s proximity to the splenic vessels and the lack of a definitive benign diagnosis.

An intraoperative frozen section analysis revealed aggregates of monomorphic epithelial cells arranged in trabeculae and micronodules, raising suspicion for neuroendocrine neoplasia or hyperplasia. However, a definitive histopathological examination showed the foci of chronic sclerosing pancreatitis with acinar atrophy, ductal hyperplasia, and islet hyperplasia. Importantly, three peripancreatic lymph nodes were examined and found to be uninvolved.

Following the pancreatectomy, the patient had an uneventful recovery. Steroid therapy was not initiated, particularly to avoid the potential risk of diabetes mellitus. At subsequent follow-up visits, the patient remained symptom-free, with no recurrence of pancreatitis or related complications.

## 3. Discussion

AIP is a rare and diagnostically challenging condition. Since its first case reports nearly 30 years ago, the need to distinguish AIP from other pancreatic diseases has been widely recognized. Due to the overlapping clinical symptoms and radiological features, AIP can be misdiagnosed as pancreatic adenocarcinoma, leading to unnecessary pancreatectomies. In response, numerous publications have proposed diagnostic criteria to mitigate this issue.

In 2012, Onda et al. reported the first histologically proven case of AIP misdiagnosed as a PanNEN. The case involved a 53-year-old man with a hypoechoic pancreatic mass detected via ultrasonography during a routine check-up. Subsequent imaging, including CT, MRI, ERCP, and EUS, led to the suspicion of a non-functioning PanNEN or malignant neoplasm. Following enucleation, a histological analysis confirmed type 1 AIP as the final diagnosis [29].

In 2023, Sharma et al. described the first case of chronic pancreatitis misdiagnosed as a PanNEN due to focal Ga-68 DOTATATE PET-CT uptake. This involved a 20-year-old male presenting with abdominal pain and a 3 cm soft tissue lesion in the pancreas, identified on abdominal CT. EUS-guided fine needle aspiration (FNA) revealed round cells with mild nuclear pleomorphism and cytoplasmic content suggestive of a PanNEN. Ga-68 DOTATATE PET-CT showed increased tracer uptake in the lesion, further supporting the initial PanNEN diagnosis [30]. To our knowledge, the case we present is the second involving a young male patient investigated for recurrent acute pancreatitis and diagnosed with a PanNEN based on Ga-68 DOTATATE PET-CT findings. In our case, the EUS findings were limited, and the Ga-68 DOTATATE PET-CT tracer uptake contributed to a misdiagnosis of a PanNEN, initially corroborated by intraoperative frozen section histology. While Ga-68 DOTATATE PET-CT remains a pivotal tool in a PanNEN diagnosis, with a sensitivity and specificity of approximately 93% [31], false positives can occur, as this case illustrates.

Sharma et al. hypothesized that in chronic pancreatitis, including AIP, neuroendocrine hyperplasia and clusters of residual islet cells within fibrotic areas might explain the increased Ga-68 DOTATATE uptake. These histological changes could also lead to cytological misinterpretations consistent with PanNENs.

Another plausible explanation in our case is that during AIP, the diffuse endocrine system, particularly islet cells, may undergo reactive changes in response to chronic inflammation, resulting in false-positive Ga-68 DOTATATE PET-CT findings. This hypothesis suggests that autoimmune inflammation in AIP may affect both exocrine and endocrine pancreatic components, leading to complex histopathological and imaging findings.

These cases underscore that AIP can be misdiagnosed not only as pancreatic adenocarcinoma but also as PanNENs. The diagnostic complexity is further heightened by reports of AIP coexisting with PanNEN [32] and instances where PanNEN mimics AIP, such as in cases presenting with diffuse pancreatic enlargement [33]. This raises intriguing questions about a potential interplay between these conditions. For example, could an indolent neuroendocrine tumor act as a chronic immune stimulus, triggering an autoimmune response and the subsequent development of AIP? Alternatively, could the autoimmune inflammation characteristic of AIP promote an environment conducive to neuroendocrine neoplasia? Although not yet widely explored, the relationship between neuroendocrine neoplasms and autoimmune reactions could provide valuable insights into cases where these conditions coexist or are misdiagnosed as one another.

## 4. Conclusions

In conclusion, despite the advances in diagnostic criteria and increased clinician awareness, AIP remains a difficult-to-diagnose condition. No single diagnostic test is sufficient on its own. Instead, an integrative approach using all the available clinical, radiological, and histological tools is essential to achieve an accurate diagnosis and avoid unnecessary interventions.

## Figures and Tables

**Figure 1 ijms-26-01536-f001:**
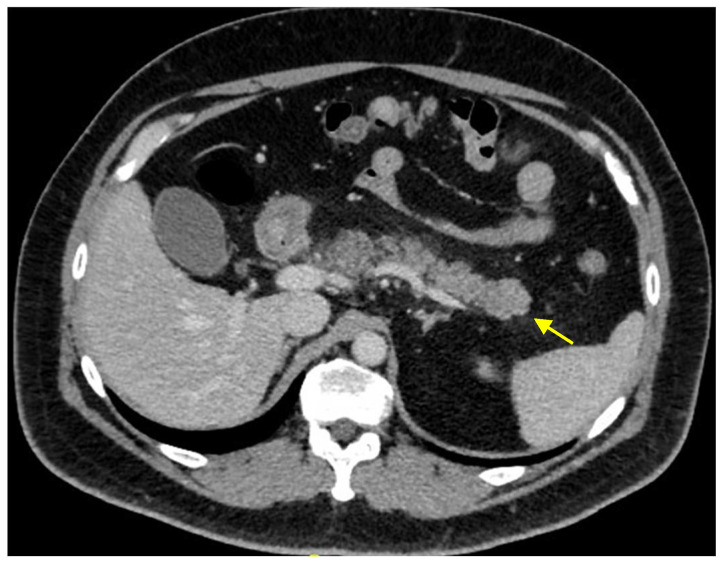
A CT scan performed with iodinated contrast media demonstrated a lesion in the distal pancreatic tail, measuring 23 mm × 18 mm (yellow arrow). The lesion appeared isodense to the surrounding pancreatic parenchyma across all contrast phases. It exhibited a compact glandular component with polylobulated margins. The spleen, liver, and surrounding retroperitoneal structures showed no abnormalities.

**Figure 2 ijms-26-01536-f002:**
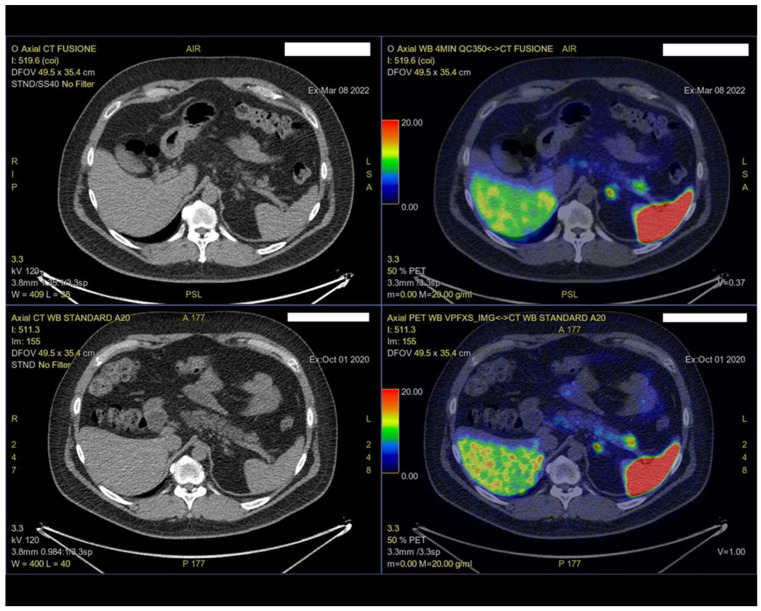
Increased tracer uptake in the pancreatic tail lesion was observed on Ga-68 DOTATATE PET-CT, suggestive of somatostatin receptor expression. The Ga-68 DOTATATE PET-CT was performed using an Omni Legend PET/CT scanner (GE HealthCare, Chicago, IL, USA).

**Figure 3 ijms-26-01536-f003:**
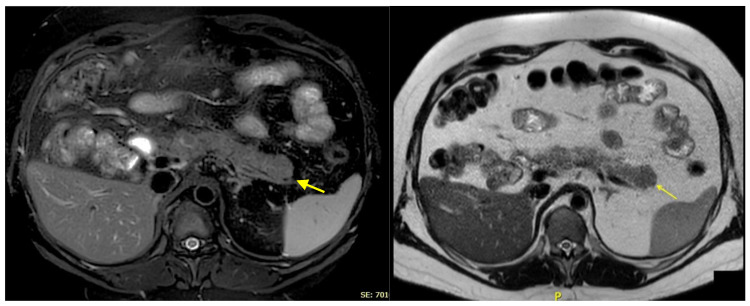
A pseudonodular alteration was identified in the pancreatic tail, measuring approximately 25 mm. On T1-weighted post-contrast MRI sequences, the lesion appeared isointense relative to the surrounding pancreatic tissue, while on T2-weighted sequences, it was slightly hypointense (yellow arrows). The MRI was performed with a 1.5T magnet (Ingenia, Philips Healthcare, Amsterdam, The Netherlands). P = posterior.

## Data Availability

The data supporting the findings of this study are available within the manuscript. Any additional data can be provided upon reasonable request.

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
