# Peer review of "Autoimmune Pancreatitis Mimicking a Pancreatic Neuroendocrine Tumor: A Case Report with a Literature Review"

_ijms, 2025, doi:10.3390/ijms26041536_

Round 1
Reviewer 1 Report
Comments and Suggestions for Authors
I read with great interest this case report, which corresponds to a difficult clinical routine, even in experienced centres. I have a few comments
You said : "Computed tomography (CT) showed peripancreatic fat stranding, multiple reactive lymph nodes, and thickening of the left anterior parenchymal fascia, consistent with acute pancreatitis"
Please details the location of the lymph nodes
You said : "At a one-month follow-up CT scan, a more compact glandular component with polylobulated margins, measuring 23 × 18 mm, was identified at the distal pancreatic tail, without detectable Doppler signal"
I don't understand what the Doppler signal has to do with it. In addition, more detail of the CT scan is needed (type of protocol and contrast media used) and add information concerning enhancement pattern of the lésion compared to sourronding parenchyma.
regarding MR, what were the other elements that suggested the diagnosis of NET ? e.g. diffusion, enhancement ?
Biopsy (FNB) report is called “inconclusive.” It is important to detail why it was inconclusive. Was it a problem of quantity of tissue, contamination of the specimen, difficulty of the procedure, or difficulty in cytologic interpretation?
And why the EUS-guided biopsy was not repeated despite the inconclusive result).
Please add a sentence to justify the surgical procedure, since distal pancraetectomy without splenectomy is actually an accepted technique, also for PNET.
it would be interesting to better show the iconography of diagnostic tests, including CT and EUS, and if it is possible macroscopic and microscopic pathology
Author Response
“I read with great interest this case report, which corresponds to a difficult clinical routine, even in experienced centres. “
We thank the reviewer for his appreciation.
"Computed tomography (CT) showed peripancreatic fat stranding, multiple reactive lymph nodes, and thickening of the left anterior parenchymal fascia, consistent with acute pancreatitis"
Please details the location of the lymph nodes
Thank you for your observation. In the revised manuscript, we have clarified that the reactive lymph nodes were located in the peripancreatic region, particularly along the splenic artery and within the splenic hilum (see page 3).
"At a one-month follow-up CT scan, a more compact glandular component with polylobulated margins, measuring 23 × 18 mm, was identified at the distal pancreatic tail, without detectable Doppler signal"
I don't understand what the Doppler signal has to do with it. In addition, more detail of the CT scan is needed (type of protocol and contrast media used) and add information concerning enhancement pattern of the lésion compared to sourronding parenchyma.
We appreciate your comment and have corrected the manuscript to remove the unnecessary reference to Doppler signal. Additionally, we have provided more comprehensive details regarding the CT scan, including the use of a contrast-enhanced multiphase protocol and the type of iodinated contrast media. Furthermore, we elaborated on the lesion's enhancement pattern, describing it as hypodense relative to the surrounding pancreatic parenchyma during all phases (see page 3).
Regarding MR, what were the other elements that suggested the diagnosis of NET ? e.g. diffusion, enhancement?
The revised manuscript includes a detailed description of the MRI findings. The lesion exhibited minimal hypointensity on T2-weighted sequences, with no diffusion restriction and mild arterial-phase enhancement. These features, combined with the clinical context, and mainly based on Gallium-68 DOTATATE positron emission tomography-CT results initially suggested a diagnosis of NET (see page 4).
Comment 4: Biopsy (FNB) report is called “inconclusive.” It is important to detail why it was inconclusive. Was it a problem of quantity of tissue, contamination of the specimen, difficulty of the procedure, or difficulty in cytologic interpretation? And why the EUS-guided biopsy was not repeated despite the inconclusive result).
We have expanded the explanation of the inconclusive FNB in the revised text. The specimen was deemed insufficient for diagnosis due to limited cellularity and difficulty in cytologic interpretation. The decision not to repeat the procedure was influenced by the patient’s preference to avoid further invasive procedures and the high clinical and radiological suspicion of NET, which prioritized surgical intervention. This has been clarified to ensure readers understand the limitations encountered during the diagnostic process (see page 4).
Comment 5: Please add a sentence to justify the surgical procedure, since distal pancreatectomy without splenectomy is actually an accepted technique, also for PNET.
We have added a justification for performing distal pancreatectomy with splenectomy. The lesion’s proximity to the splenic vessels and the need for en bloc resection to ensure clear surgical margins warranted this approach; indeed neither imaging techniques nor biopsy was conclusive for a neuroendocrine tumor or other benign disease. However, we also acknowledged in the text that distal pancreatectomy without splenectomy can be a valid alternative in selected cases (see page 5).
Reviewer 2 Report
Comments and Suggestions for Authors
Dear Authors,
you present an intriguing case study of autoimmune pancreatitis that was initially misdiagnosed as a pancreatic neuroendocrine tumour.
This article underscores the diagnostic challenges posed by autoimmune pancreatitis, given its non-specific symptoms that can mimic other pancreatic disorders.
I don't have many observations except that one could cite: Okazaki K, et al. J Gastroenterol. While not drastically altering existing guidelines, Okazaki K provides a more recent overview of this condition and could be a valuable addition to the discussion.
Missing Information: The article lacks essential sections such as "Author contributions," "Funding," "Institutional review board statement," "Informed consent," "Data availability," and "Conflict of interest."
In this regard, I believe that the approval of an ethics committee is paramount.
I believe that article ijms-3401389 is particularly interesting due to the topic it addresses. Autoimmune pancreatitis (AIP) is a rare and insidious form of chronic pancreatitis that can closely mimic pancreatic cancer, significantly complicating diagnostic efforts due to overlapping clinical and radiological presentations. Differential diagnosis is therefore as crucial as it is complex. The approach suggested in this case report is multidisciplinary. It involves not only the clinician but also the radiologist, nuclear medicine physician, laboratory physician, and surgeon. The take home message could be that in this disease, albeit it's rarity, collaboration among specialists and a holistic approach to the patient are paramount to reach the correct diagnosis.
For these reasons, I believe the article should be considered for publication.
Author Response
Reviewer wrote:
Dear Authors,
you present an intriguing case study of autoimmune pancreatitis that was initially misdiagnosed as a pancreatic neuroendocrine tumor.
This article underscores the diagnostic challenges posed by autoimmune pancreatitis, given its non-specific symptoms that can mimic other pancreatic disorders.
Thank you for your thorough review and positive remarks regarding our case report. We are grateful for your recognition of the diagnostic challenges posed by autoimmune pancreatitis (AIP) and your acknowledgment of the multidisciplinary approach emphasized in our study.
I don't have many observations except that one could cite: Okazaki K, et al. J Gastroenterol. While not drastically altering existing guidelines, Okazaki K provides a more recent overview of this condition and could be a valuable addition to the discussion.
We appreciate your suggestion to include the work of Okazaki K et al. from J Gastroenterol. We have incorporated these citations (ref #7 an 8) into the discussion to enrich the context and provide readers with an additional resource for understanding the condition.
Missing Information: The article lacks essential sections such as "Author contributions," "Funding," "Institutional review board statement," "Informed consent," "Data availability," and "Conflict of interest."
In this regard, I believe that the approval of an ethics committee is paramount.
We have addressed the missing sections in the revised manuscript.
I believe that article ijms-3401389 is particularly interesting due to the topic it addresses. Autoimmune pancreatitis (AIP) is a rare and insidious form of chronic pancreatitis that can closely mimic pancreatic cancer, significantly complicating diagnostic efforts due to overlapping clinical and radiological presentations. Differential diagnosis is therefore as crucial as it is complex. The approach suggested in this case report is multidisciplinary. It involves not only the clinician but also the radiologist, nuclear medicine physician, laboratory physician, and surgeon. The take home message could be that in this disease, albeit it's rarity, collaboration among specialists and a holistic approach to the patient are paramount to reach the correct diagnosis.
For these reasons, I believe the article should be considered for publication.
We appreciate your emphasis on the importance of a multidisciplinary approach and holistic patient care. To highlight this in the manuscript, we have enhanced the conclusion by explicitly stating that collaboration among clinicians, radiologists, nuclear medicine specialists, laboratory physicians, and surgeons is pivotal in navigating the diagnostic complexities of AIP. This aligns well with the take-home message you proposed.
Round 2
Reviewer 1 Report
Comments and Suggestions for Authors
I would like to thank the authors for their comprehensive and precise answers to the points I had found improvable.
I only believe that the iconography needs to be improved a little, with the addition of CT imaging and further MRI sequences.
Author Response
We sincerely thank the reviewer for acknowledging the effort we put into addressing the previous comments. We appreciate the suggestion regarding the iconography and have taken steps to improve it.
"I only believe that the iconography needs to be improved a little, with the addition of CT imaging and further MRI sequences."
Response: We have added the requested CT imaging as well as further MRI sequences to provide a more comprehensive visualization of the lesion. These additions have been incorporated into the revised manuscript and are referenced accordingly in the updated figure legends.
Thank you once again for your valuable feedback, which has significantly enhanced the quality of our work.